# Manipulations of MeCP2 in glutamatergic neurons highlight their contributions to Rett and other neurological disorders

Xiangling Meng[1,2], Wei Wang[2,3], Hui Lu[2,3], Ling-jie He[2,3,4], Wu Chen[1,5], Eugene S Chao[1,5], Marta L Fiorotto[6], Bin Tang[2,7], Jose A Herrera[2,8], Michelle L Seymour[9,10], Jeffrey L Neul[2†], Fred A Pereira[9,10,11], Jianrong Tang[2,7], Mingshan Xue[1,5], Huda Y Zoghbi[1,2,3,4]*

[1]Department of Neuroscience, Baylor College of Medicine, Houston, United States; [2]Jan and Dan Duncan Neurological Research Institute, Texas Children's Hospital, Houston, United States; [3]Department of Molecular and Human Genetics, Baylor College of Medicine, Houston, United States; [4]Howard Hughes Medical Institute, Baylor College of Medicine, Houston, United States; [5]The Cain Foundation Laboratories, Jan and Dan Duncan Neurological Research Institute, Texas Children's Hospital, Houston, United States; [6]Children's Nutrition Research Center, Department of Pediatrics, Baylor College of Medicine, Houston, United States; [7]Department of Pediatrics, Baylor College of Medicine, Houston, United States; [8]Interdepartmental Program in Translational Biology and Molecular Medicine, Baylor College of Medicine, Houston, United States; [9]Huffington Center on Aging, Baylor College of Medicine, Houston, United States; [10]Department of Molecular and Cellular Biology, Baylor College of Medicine, Houston, United States; [11]Bobby R Alford Department of Otolaryngology - Head and Neck Surgery, Baylor College of Medicine, Houston, United States

*For correspondence: hzoghbi@bcm.edu

Present address: †Department of Neurosciences, University of California, San Diego, United States

**Abstract** Many postnatal onset neurological disorders such as autism spectrum disorders (ASDs) and intellectual disability are thought to arise largely from disruption of excitatory/inhibitory homeostasis. Although mouse models of Rett syndrome (RTT), a postnatal neurological disorder caused by loss-of-function mutations in *MECP2*, display impaired excitatory neurotransmission, the RTT phenotype can be largely reproduced in mice simply by removing MeCP2 from inhibitory GABAergic neurons. To determine what role excitatory signaling impairment might play in RTT pathogenesis, we generated conditional mouse models with *Mecp2* either removed from or expressed solely in glutamatergic neurons. MeCP2 deficiency in glutamatergic neurons leads to early lethality, obesity, tremor, altered anxiety-like behaviors, and impaired acoustic startle response, which is distinct from the phenotype of mice lacking MeCP2 only in inhibitory neurons. These findings reveal a role for excitatory signaling impairment in specific neurobehavioral abnormalities shared by RTT and other postnatal neurological disorders.

**eLife digest** Rett syndrome is a childhood brain disorder that mainly affects girls and causes symptoms including anxiety, tremors, uncoordinated movements and breathing difficulties. Rett syndrome is caused by mutations in a gene called *MECP2*, which is found on the X chromosome. Males with *MECP2* mutations are rare but have more severe symptoms and die young. Many researchers who study Rett syndrome use mice as a model of the disorder. In particular, male mice with the mouse equivalent of the human *MECP2* gene switched off in every cell in the body (also known as *Mecp2*-null mice) show many of the features of Rett syndrome and die at a young age.

The *MECP2* gene is important for healthy brain activity. The brain contains two major types of neurons: excitatory neurons, which encourage other neurons to be active; and inhibitory neurons, which stop or dampen the activity of other neurons. In 2010, researchers reported that mice lacking *Mecp2* in only their inhibitory neurons develop most of the same problems as those mice with no *Mecp2* at all.

Now, Meng et al. – who include two researchers involved in the 2010 study – have asked how deleting or activating *Mecp2* only in excitatory neurons of mice affects Rett-syndrome-like symptoms. The experiments showed that male mice without *Mecp2* in their excitatory neurons develop tremors, anxiety-like behaviors, abnormal seizure-like brain activity and severe obesity; these mice also die earlier than normal mice. Female mice lacking *Mecp2* in half of their excitatory neurons (because the gene is on the X chromosome) were less affected than the males, and had normal life spans. These symptoms are different from those seen in mice missing *Mecp2* only in inhibitory neurons.

Meng et al. also found that if *Mecp2* was switched on only in excitatory neurons of female mice (which are a model of human Rett syndrome patients) the mice were almost completely normal. In male mice (which show more severe symptoms), activating *Mecp2* in only the excitatory neurons reduced the anxiety and tremors. These findings suggest that impaired excitatory neurons may be responsible for specific symptoms such as anxiety and tremors amongst other Rett-syndrome-like features.

The next challenge is to explore how the loss of *Mecp2* changes the activity of excitatory neurons in different brain regions. Further studies could also investigate if drugs that improve the activity of excitatory neurons can be used to treat Rett syndrome patients. Finally, in a related study, Ure et al. asked if activating *Mecp2* in inhibitory neurons in otherwise *Mecp2*-null mice was enough to prevent some of their Rett syndrome-like symptoms.

## Introduction

A number of postnatal neurological disorders such as autism spectrum disorders (ASDs) and certain types of intellectual disability are thought to involve a disturbance in excitatory/inhibitory homeostasis (*Nelson and Valakh, 2015*; *Rubenstein and Merzenich, 2003*). For example, mouse models of Rett syndrome (RTT), a severe postnatal onset neurological disorder, show diminished excitatory drive and overall reduced cortical microcircuit activity (*Dani et al., 2005*; *Wood et al., 2009*). Conversely, the *Met* conditional deletion mouse (a model of nonsyndromic, sporadic autism) shows increased excitatory synaptic connectivity from cortical layer 2/3 to layer 5B corticostriatal neurons (*Qiu et al., 2011*). Certainly the prevalence of epilepsy as a comorbidity with ASDs and intellectual disability disorders would suggest a tendency toward hyperexcitation, but both excitatory and inhibitory neuronal populations are affected in these disorders. It is difficult to discern the contributions of different neuronal groups to pathogenesis, however, in part because genes driving disorders such as RTT, Fragile X syndrome, Angelman syndrome, and tuberous sclerosis (*MECP2*, *FMRP*, *UBE3A*, and *TSC1* and *TSC2*, respectively) participate in fundamental cellular processes in multiple cell types (*Crino, 2013*; *Darnell et al., 2011*; *Lyst and Bird, 2015*; *Mabb et al., 2011*), and in part because network alterations always induce compensatory changes in the circuit (*Turrigiano, 2011*).

One way to circumvent this challenge is to examine the effects of such genes on specific neuronal types. This is the approach we and others have taken to understand the contribution of various

neuronal types to the complex, wide-ranging phenotype of RTT, which is caused by loss-of-function mutations in the ubiquitously expressed *MECP2* (methyl CpG-binding protein 2) (*Amir et al., 1999*; *Trappe et al., 2001*). Removing *Mecp2* from SIM-1 expressing hypothalamic neurons, for example, results in hyperphagia, obesity and aggression (*Fyffe et al., 2008*), whereas deficiency of MeCP2 in somatostatin-positive neurons causes seizures and stereotypies (*Ito-Ishida et al., 2015*). The most surprising result, however, given early studies suggesting RTT involves impairment in excitatory signaling (*Chao et al., 2007*; *Chapleau et al., 2009*; *Dani et al., 2005*; *Marchetto et al., 2010*), came from conditional knockout of *Mecp2* in GABAergic neurons. These GABAergic conditional knockout mice reproduce most of the features observed in the *Mecp2*-null mice: premature lethality, stereotyped forepaw motions and other repetitive behaviors, abnormal social interaction, learning and memory deficits, hypoactivity, ataxia, and hindlimb clasping (*Chao et al., 2010*). The only features of the *Mecp2*-null mice that did not develop in the GABAergic knockouts were tremor and anxiety-like behaviors. What role, then, do the excitatory signaling impairments observed in Rett patients and previous mouse models play in RTT pathogenesis?

To answer this question, we characterized mice with *Mecp2* deleted solely in excitatory glutamatergic neurons. We further examined null mice with *Mecp2* re-expressed only in the glutamatergic neurons to determine which behaviors healthy excitatory neurons were sufficient to rescue. We found that loss of MeCP2 in glutamatergic neurons results in neurological deficits that are quite distinct from those revealed upon loss of MeCP2 in GABAergic neurons.

## Results

### *Vglut2*-Cre mediated deletion and re-expression of MeCP2 in glutamatergic neurons

To target excitatory neurons specifically, we utilized an *Slc17a6*-Cre (also known as *vesicular glutamate transporter 2*-Cre, referred to as *Vglut2*-Cre henceforth) mouse line to express Cre recombinase in the majority of excitatory neurons (*Vong et al., 2011*). *Vglut2* is the dominant glutamate vesicular transporter at embryonic and early postnatal stages: by P14 it is mostly replaced by *Vglut1* in the hippocampus, cortex, and cerebellum (*Boulland et al., 2004*; *Fremeau et al., 2004*; *Miyazaki et al., 2003*), which ensures that any Cre activity driven by *Vglut2* will have a broad effect on excitatory neurons. Indeed, characterization of *Vglut2*-Cre expression using Rosa-tdTomato reporter mice revealed that the majority of CamKII-positive glutamatergic neurons in the cortex and hippocampal CA1 region expressed Cre (88% and 98%, respectively; *Figure 1A and B*). 99% of Cre-expressing cells were CamKII-positive (*Figure 1A and B*), indicating that Cre is specifically expressed in excitatory neurons. To conditionally delete *Mecp2* in glutamatergic neurons we mated female mice carrying a conditional *Mecp2* allele flanked by loxP sites (*Guy et al., 2001*) with male *Vglut2*-Cre$^{+/-}$ mice and generated *Mecp2* conditional knockout mice (*Mecp2*$^{flox+/y}$;*Vglut2*-Cre$^{+/-}$: CKO) along with their littermate controls (*Mecp2*$^{flox+/y}$: Flox, wild type: WT, *Vglut2*-Cre$^{+/-}$: Cre). Immunofluorescence staining of MeCP2 in the CKO mice demonstrated a clear loss of the protein in areas primarily composed of excitatory neurons, such as the cerebral cortex, thalamus, and hippocampal CA1 through CA3 regions (*Figure 1C*).

To restore MeCP2 function in glutamatergic neurons, we bred *Vglut2*-Cre$^{+/-}$ male mice to females carrying a *Mecp2* conditional rescue allele with a floxed STOP cassette (*Mecp2*$^{LSL/+}$) (*Guy et al., 2007*). Male F1 *Mecp2*$^{LSL/y}$ mice (referred to as 'stop-null') express a truncated, non-functioning version of MeCP2 and phenotypically resemble the constitutive null mutants (*Guy et al., 2007*). Male *Mecp2*$^{LSL/y}$;*Vglut2*-Cre$^{+/-}$ mice (referred to as 'conditional rescue' or 'C-rescue') re-express MeCP2 only in glutamatergic neurons, where Cre excises the STOP cassette. Consistent with published data, there was no MeCP2 expression in stop-null mice (*Guy et al., 2007*), but its expression was restored in regions rich in excitatory neurons in the male C-rescue mice (*Figure 1D*). Specifically, 95% of CamKII-positive glutamatergic neurons in the cortex of male C-rescue mice expressed MeCP2 while all MeCP2-positive neurons were labeled by CamKII (*Figure 1E and F*), indicating successful restoration of MeCP2 specifically in glutamatergic neurons.

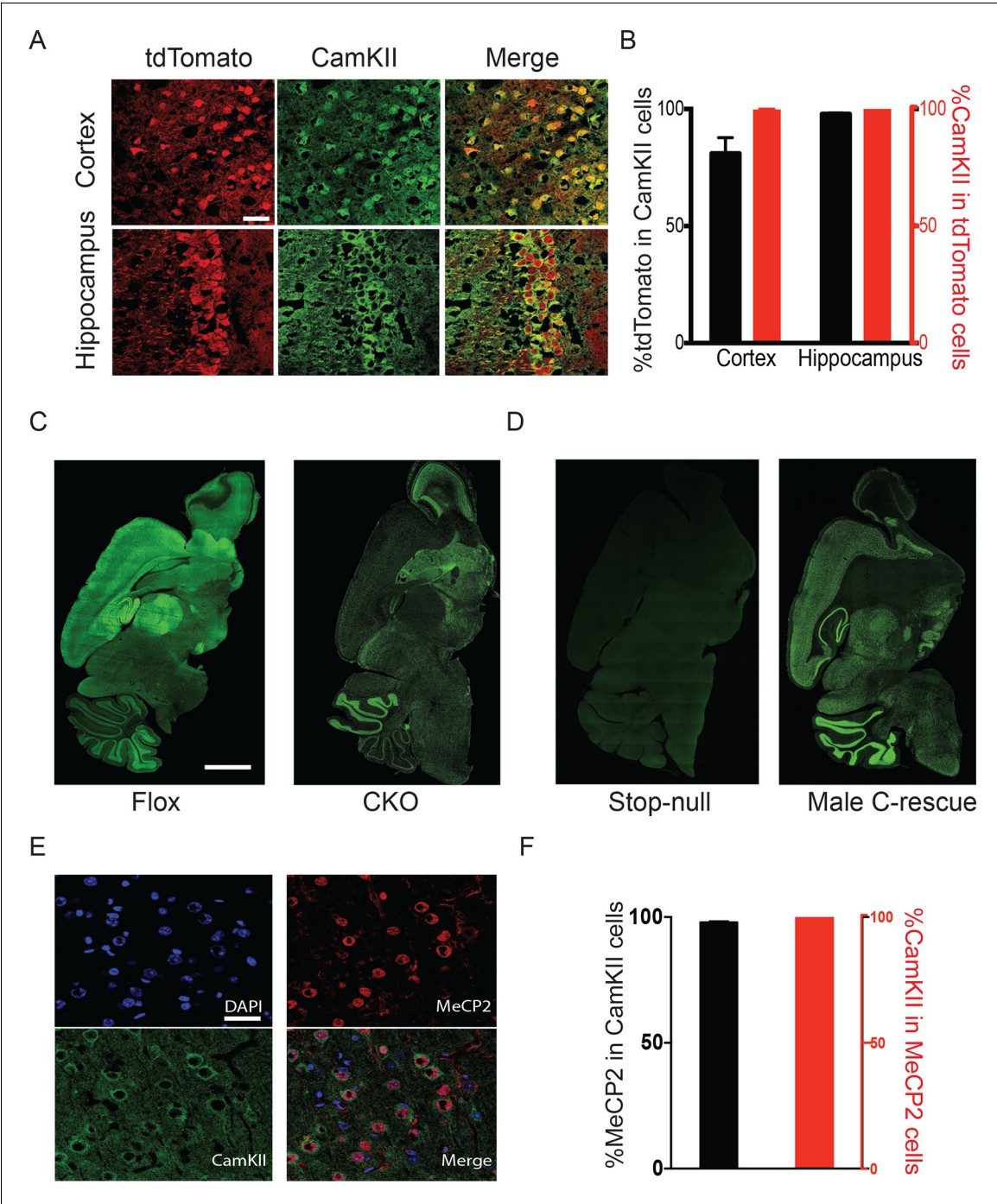

**Figure 1.** *Mecp2* was either deleted or restored specifically in glutamatergic neurons. (**A**) *Vglut2*-Cre expression was assayed by colocalization of the reporter tdTomato and CamKII in 6-week-old *Vglut2*-Cre$^{+/-}$;*Rosa26*$^{tdTomato}$ male mice. Scale bar, 30 μm. (**B**) Quantification of images in (**A**) showing the percentage of tdTomato positive cells in total CamKII-immunostaining cells (black) and the percentage of CamKII-immunostaining cells in total tdTomato positive cells (red, n = 3 mice, 14 sections). (**C,D**) Representative images showing MeCP2 expression in the brain of Flox and CKO mice (**C**), as well as stop-null and male C-rescue mice (**D**). Scale bar, 2 mm (n = 3 mice per genotype). (**E**) Fluorescence images of male C-rescue cortex stained for nucleus (4′,6-diamidino-2-phenylindole, DAPI), MeCP2 and CamKII. Scale bar, 30 μm. (**F**) Quantification of images in (**E**) showing the percentage of MeCP2 positive cells in total CamKII-immunostaining cells (black), and the percentage of CamKII-immunostaining cells in total MeCP2 positive cells (red, n = 3 mice, 18 sections).

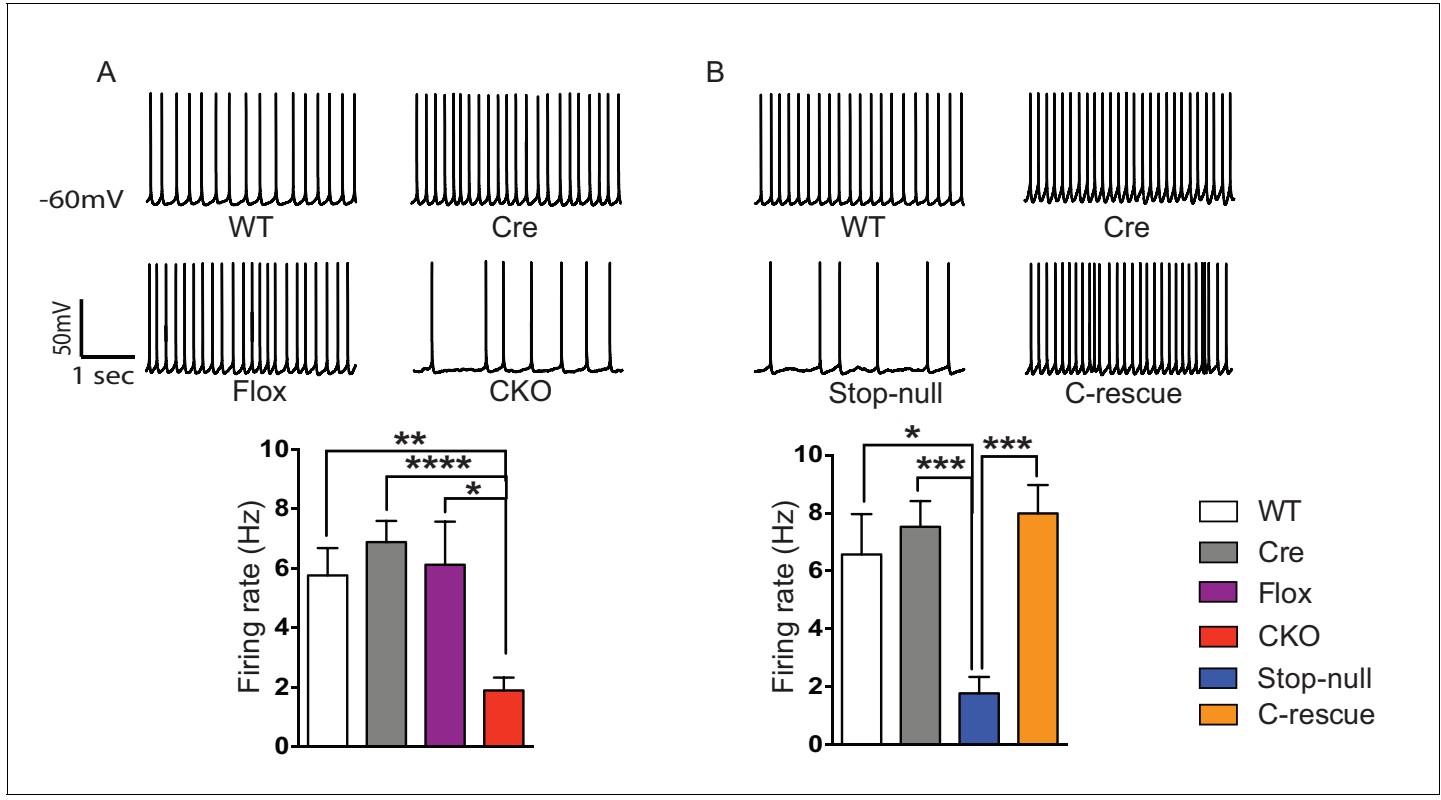

**Figure 2.** Reduced cortical activity was induced by conditional deletion of *Mecp2* in glutamatergic neurons but rescued by conditional restoration of MeCP2. (**A**) Top: Sample traces of spontaneous firing of Layer V neurons from CKO and control mice. Bottom: Average firing rate of four genotypes of mice. (**B**) Top: Sample traces of spontaneous firing of Layer V neurons from male C-rescue mice and its counterparts. Bottom: Quantification of spontaneous firing rate. n = 3–4 mice and 12–15 cells per genotype. Data are presented as mean ± SEM. *p<0.05; **p<0.01; ***p<0.001; ****p<0.0001; by one-way ANOVA.

The following figure supplements are available for figure 2:

**Figure supplement 1.** Layer V pyramidal neurons in the CKO mice received less excitatory and inhibitory input.

**Figure supplement 2.** CKO mice displayed reduced mEPSC frequency and normal mIPSCs.

**Figure supplement 3.** Restoration of MeCP2 in glutamatergic neurons normalized reduced mEPSC frequency in stop-null mice.

## Expression of MeCP2 in glutamatergic neurons plays an essential role in maintaining excitatory neuron activity

To determine the functional importance of MeCP2 in glutamatergic neurons, we studied the effect of loss of MeCP2 on cortical layer V pyramidal neurons in the somatosensory cortex. Whole-cell patch-clamp recordings from these neurons in 6- to 8-week old mice revealed reduced spontaneous action potential firing in CKO mice (*Figure 2A*) similar to that observed in the *Mecp2*-null mutants (*Dani et al., 2005*). In the presence of synaptic transmission blockers, neurons in CKO mice showed similar firing rates in response to injected currents as WT neurons, indicating normal intrinsic excitabilities (*Figure 2—figure supplement 1C*). To explore the cause of reduced spontaneous activity, we focused on the total excitatory and inhibitory drive onto layer V pyramidal neurons. In the presence of ongoing network activity within the slices, we recorded spontaneous excitatory postsynaptic currents (sEPSCs) by voltage-clamping the cell membrane at the measured chloride-reversal potential (WT: −70.8 ± 0.4 mV; Cre: −69.1 ± 0.3 mV; Flox: −70.7 ± 0.5 mV; CKO: −69.9 ± 0.6 mV), while spontaneous inhibitory postsynaptic currents (sIPSCs) were recorded by holding the cell at the measured cation-reversal potential (WT: 14.3 ± 0.4 mV; Cre: 12.9 ± 0.4 mV; Flox: 14.3 ± 0.8 mV; CKO: 12.7 ± 0.2 mV). Interestingly, both excitatory and inhibitory synaptic charges were reduced in CKO

mouse neurons compared to controls (*Figure 2—figure supplement 1A and B*), indicating that the observed reduction in cortical activity is most likely caused by reduced excitatory input to layer V neurons. To determine if the change in sEPSCs is caused by an alteration in quantal responses (miniature EPSCs, mEPSCs) or spike-driven EPSCs, we recorded α-amino-3-hydroxy-5-methyl-4-isoxazole-propionic acid receptor (AMPAR)-dependent mEPSCs. The frequency of mEPSCs recorded from CKO mice was significantly decreased when compared to WT mice and showed a trend of reduction although not significant when compared with Cre or Flox mice (*Figure 2—figure supplement 2A*). The amplitude of mEPSCs and both the amplitude and frequency of miniature IPSCs (mIPSCs) did not differ significantly across genotypes (*Figure 2—figure supplement 2A and B*). These data suggest that the weakened spike-driven excitatory input to layer V pyramidal neurons largely accounts for the reduced firing rate in CKO mice.

To determine if restoration of MeCP2 function in glutamatergic neurons can improve the reduced cortical activity in *Mecp2*-null mice, we measured spontaneous action potential firing in the male C--rescue, stop-null, and control mice. As expected, layer V pyramidal neurons in the somatosensory cortex of the stop-null mice showed a significantly reduced firing rate (*Figure 2B*), whereas restoring MeCP2 in excitatory neurons rescued the firing rate to a level similar to that observed in controls. The stop-null mice also showed reduced mEPSC frequency (*Figure 2—figure supplement 3A*) as reported in *Mecp2*-null mice (*Chao et al., 2007*), while this defect was reversed in C-rescue mice. Frequency and amplitude of mIPSCs in stop-null mice and C-rescue mice were similar to controls (*Figure 2—figure supplement 3B*). Thus, restoration of MeCP2 only in excitatory neurons is sufficient to maintain layer V pyramidal neuron activity.

## Deletion of *Mecp2* in glutamatergic neurons leads to obesity and premature death, which are improved in male C-rescue mice

To understand what features of the RTT phenotype might derive from excitatory signaling impairment, we characterized the CKO and male C-rescue mice. CKO mice died by 10 weeks of age (*Figure 3A*), similar to *Mecp2*-null mice (*Guy et al., 2001*). Interestingly, restoration of MeCP2 only in glutamatergic neurons significantly lengthened lifespan, as half of the male C-rescue mice survived for more than 46 weeks, in contrast to the ~12-week median lifespan of stop-null mice (*Figure 3B*).

The CKO mice, being an F1 hybrid of FVB and 129SvEvTac, gained significantly more weight than controls as early as 6 weeks of age and became severely obese with age (*Figure 3C*), phenocopying the *Mecp2*-null mice on a 129SvEvTac background (*Heckman et al., 2014*). The male C-rescue mice gained some weight with age, but less than littermates, and they remained underweight (*Figure 3D*). To identify the cause of weight abnormalities in both CKO and C-rescue mice, we housed mice individually in calorimetry chambers for 6 days from P25-30 to simultaneously monitor food intake, activity, and components of energy balance. After a two-day acclimation period, we assessed average food intake per day over the subsequent 3 days, accounting for initial differences in individual weights. The CKO mice consumed about 25–40% more food than controls (*Figure 3—figure supplement 1A*), and this hyperphagia was associated with significantly greater weight and fat gain (*Figure 3—figure supplement 1B and C*). Evaluation of total energy expenditure, resting metabolic rate, and total activity of the mice revealed no significant difference between CKO and controls (*Figure 3—figure supplement 1D–F*). The CKO mice had a higher respiratory exchange ratio (RER) when they had free access to food (*Figure 3—figure supplement 1G*), but normal RER during fasting (*Figure 3—figure supplement 1H*), suggesting greater lipogenesis but no impediment to fat utilization. In contrast, P25-30 male C-rescue mice consumed less food than stop-null and control mice (*Figure 3—figure supplement 2A*), which correlated with less gain of fat and weight (*Figure 3—figure supplement 2B and C*). They displayed no significant difference in total daily energy expenditure, resting metabolic rate, RER, or total activity compared with controls (*Figure 3—figure supplement 2D–H*). The obesity of CKO mice and the underweight phenotype of male C-rescue mice therefore appear to be caused by abnormal energy intake rather than altered energy expenditure.

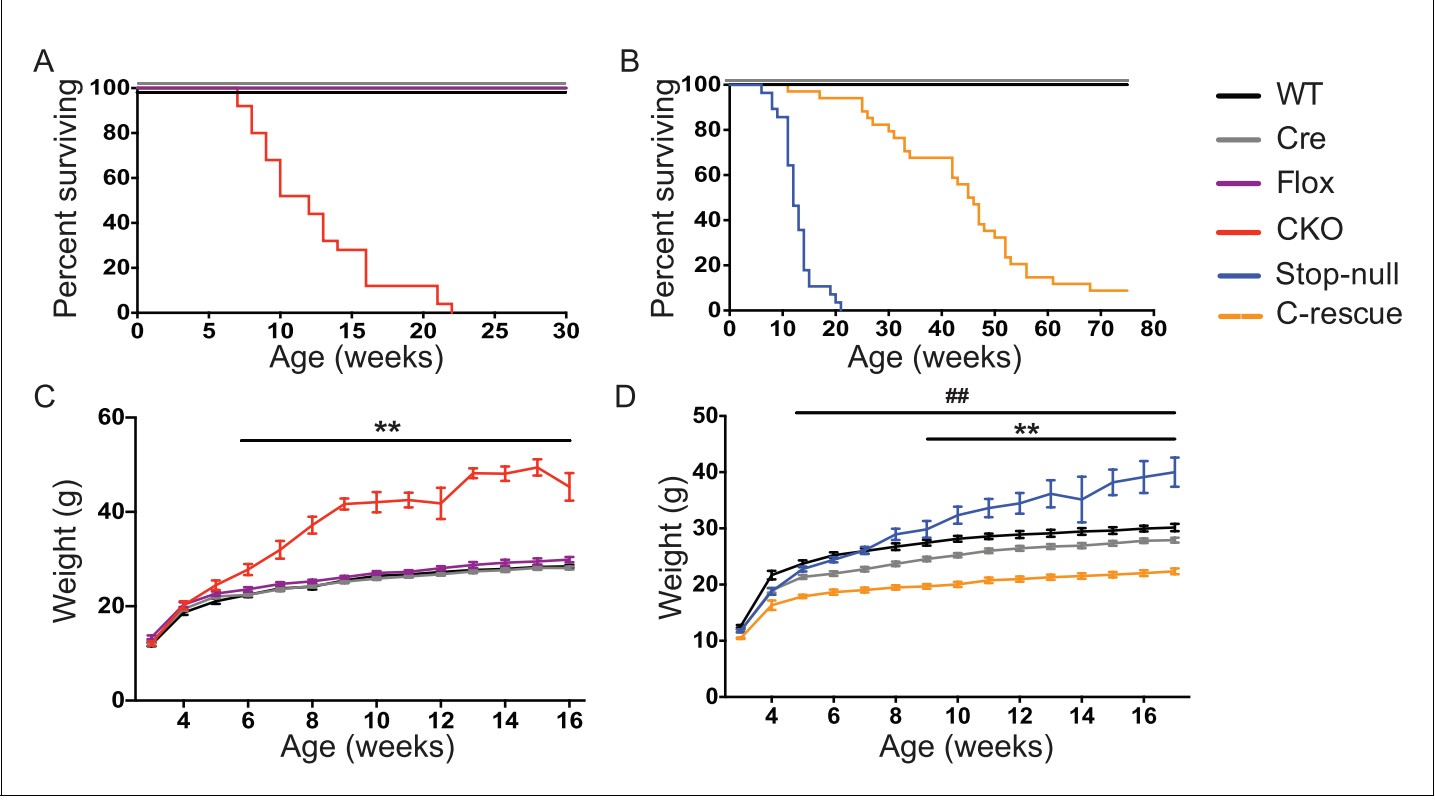

**Figure 3.** Removal of *Mecp2* from glutamatergic neurons led to early death and obesity, which were improved in C-rescue mice. (A–B) Survival curves plotted with percentage of mice alive as a function of age (A, n = 25; B, n = 28–34). (C–D) Plot of weight as a function of age (C, n = 20; D, n = 15–22). '**' indicates a statistical significant difference between CKO or stop-null and controls, while '##' indicates a statistical significant difference between C-rescue and controls. Data are presented as mean ± SEM. **, ##, p<0.01; by two-way ANOVA.

The following figure supplements are available for figure 3:

**Figure supplement 1.** CKO mice gained more weight associated with increased daily food intake.

**Figure supplement 2.** Male C-rescue mice gained less weight associated with reduced daily food intake.

## Removing MeCP2 from excitatory neurons leads to seizure-like discharges in cortical EEG recording

Epilepsy occurs in 67%–90% of individuals with RTT (*Cardoza et al., 2011*; *Cooper et al., 1998*; *Steffenburg et al., 2001*). Seizures and abnormal electroencephalography (EEG) have also been reported in *Mecp2* mutant mice (*D'Cruz et al., 2010*; *Shahbazian et al., 2002*). We found focal seizure-like spike-and-wave discharges (*Figure 4A*) in three out of eight 10-week-old CKO mice that underwent video-EEG recordings. These discharges occurred at 3.3 ± 2.1 episodes/hour (n = 3 mice) and the average duration of episodes was 4.1 ± 0.4 s (n = 27). We did not find seizure-like discharges in control mice (9 Flox, 6 Cre, and 5 WT mice). Interestingly, we observed behavioral seizures on male C-rescue mice older than 25-week-old during routine husbandary. We then conducted EEG recording on four 25- to 30-week-old male C-rescue mice, as well as two WT and two Cre mice that were at similar age. One out of those four male C-rescue mice, but no WT or Cre mice, showed spike-and-wave discharges (*Figure 4B*). They occurred at 3.3 episodes/hour and the average duration of episodes was 4.5 ± 0.8 s (n = 10). Thus, deleting MeCP2 only from glutamatergic neurons induced seizure-like abnormal EEGs, while restoring MeCP2 only in this group of neurons is not sufficient to rescue seizures.

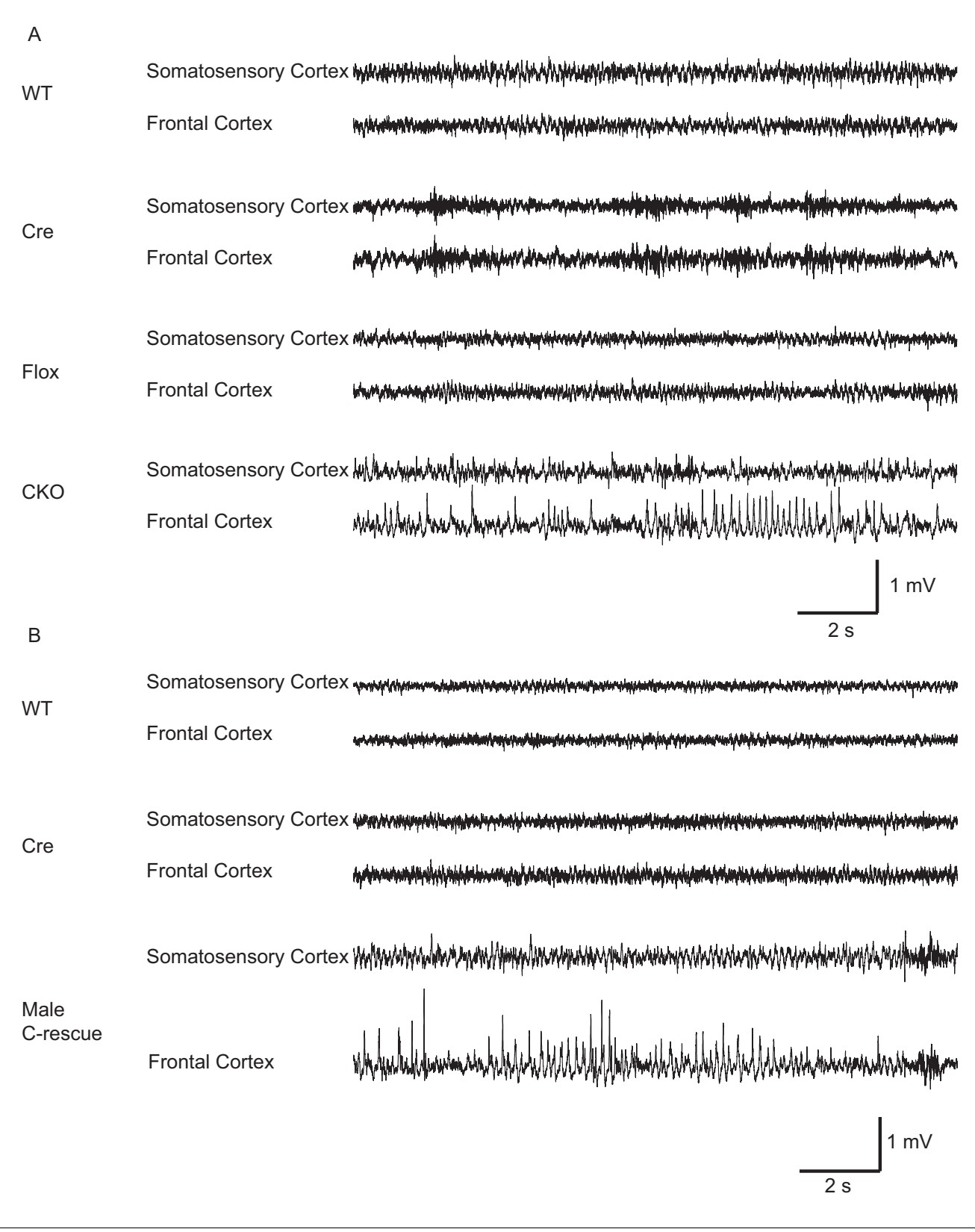

**Figure 4.** Both CKO and male C-rescue mice developed abnormal EEGs. (**A**) Representative EEG traces from the somatosensory and frontal cortices of 10-week-old WT, Cre, Flox, and CKO mice. Note the spike-and-wave discharge in the CKO mouse. (**B**) Representative EEG traces from the

*Figure 4 continued on next page*

*Figure 4 continued*

somatosensory and frontal cortices of 25- to 30-week-old WT, Cre, and male C-rescue mice. Note the spike-and-wave discharge in the male C-rescue mouse.

## Deficiency of MeCP2 in glutamatergic neurons directly mediates altered anxiety, tremor, and impaired acoustic startle response

Depletion of *Mecp2* in inhibitory GABAergic neurons reproduced most of the phenotype of the *Mecp2*-null mice except for tremor and anxiety-like behaviors (*Chao et al., 2010*). We therefore asked whether these features depend on impairment of excitatory neurons.

Our first set of behavioral tests examined anxiety-like behaviors. At 5 weeks of age, CKO mice showed similar locomotor function as controls when they traveled in the open field (*Figure 5—figure supplement 1A*). While, in the elevated plus maze test, they stayed longer in the open arms, which can indicate decreased anxiety (*Figure 5A*); on the other hand, they spent less time in the light chamber during the light/dark assay and made fewer transitions between the two chambers (*Figure 5C*), suggesting increased anxiety. These seemingly paradoxical results replicate the altered anxiety-like behaviors of *Mecp2*-null mice (*Heckman et al., 2014*; *Pelka et al., 2006*; *Stearns et al.*,

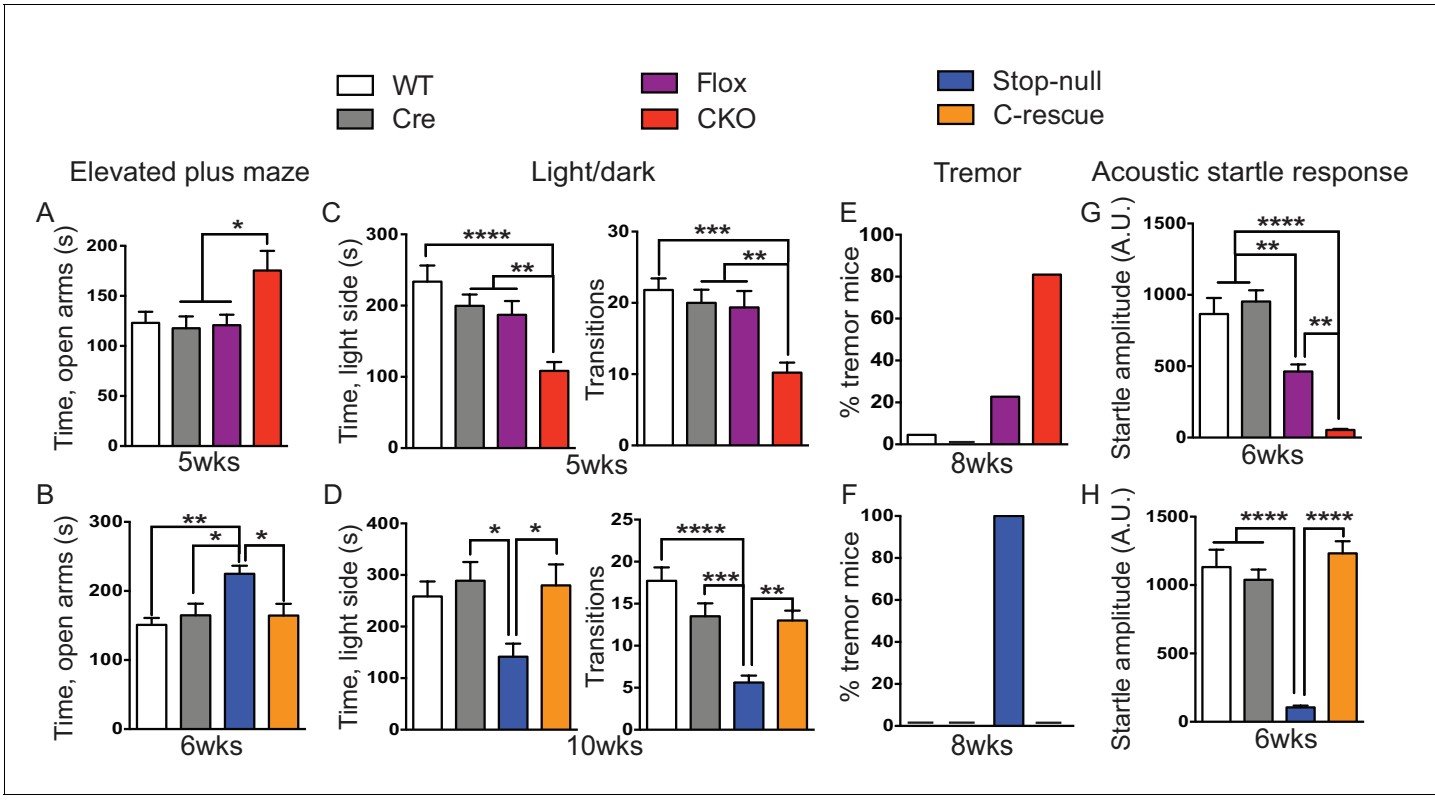

**Figure 5.** Loss of *Mecp2* in glutamatergic neurons led to altered anxiety-like behaviors, tremor, and impaired acoustic startle response, which were normalized in male C-rescue mice. (A–B) Time spent in the open arms of the elevated plus maze (A, n = 16–19; B, n = 9–14). (C–D) Average time spent in, and number of transitions into, the light chamber during 10min test in the light/dark box (C, n = 16–19; D, n = 13–16). (E–F) Percentage of mice displayed tremor at 8 weeks of age (E, n = 17–22; F, n = 18). (G–H) Mean of response to the 120 dB acoustic stimulus (G, n = 15; H, n = 14–18). Data are presented as mean ± SEM. *p<0.05; **p<0.01; ***p<0.001; ****p<0.0001; by one-way ANOVA.

The following figure supplements are available for figure 5:

**Figure supplement 1.** CKO mice have normal locomotor and hearing function.

**Figure supplement 2.** Reversal of altered anxiety-like behaviors and impaired acoustic startle response were maintained in 20-week-old mice.

*2007*). Stop-null mice displayed similarly altered anxiety-related behaviors, with increased time spent in the open arms of the elevated plus maze (*Figure 5B*) and less time in, and fewer transitions into, the light chamber (*Figure 5D*). Interestingly, restoration of MeCP2 solely in glutamatergic neurons was able to fully normalize this behavior, as the male C-rescue mice performed similarly as WT and Cre mice in those assays (*Figure 5B and D*). This reversal of altered anxiety in the male C-rescue mice was maintained to 20 weeks of age, when most stop-null mice were dead and could not be included in the experiment (*Figure 5—figure supplement 2A and B*).

At 8 weeks of age 80% of CKO mice displayed obvious tremor, which was rarely observed in control mice (*Figure 5E*). All examined stop-null mice showed tremor, while no male C-rescue mice developed a detectable tremor (*Figure 5F*).

Impaired acoustic startle response has been described in the *Mecp2*-null mice. CKO mice at 6 weeks of age also showed a diminished response to a 120 dB stimulus (*Figure 5G*). This reduction in the CKO mice is likely caused by a disturbance in the sensorimotor gating circuit in the lower brainstem (*Koch and Schnitzler, 1997*), since auditory brainstem response (ABR) measurements showed no obvious differences in hearing of the CKO mice compared to controls (*Figure 5—figure supplement 1B*). The male C-rescue mice, however, responded similarly to WT and Cre mice at 6 weeks of age, in contrast to the barely responsive stop-null mice (*Figure 5H*). At 20 weeks of age, the male C-rescue mice displayed an elevated acoustic startle response compared with controls (*Figure 5—figure supplement 2C*).

Taken together, these data imply that deficiency of MeCP2 in glutamatergic neurons underlies altered anxiety-like behaviors, tremor, and impaired acoustic startle response. This hypothesis receives further support from the failure of restoring MeCP2 expression solely in GABA-expressing neurons to rescue most of these features on a null background (see companion paper *Ure et al., 2016*).

## Deletion of MeCP2 in glutamatergic neurons leads to ataxia but spares social interaction deficits and repetitive behaviors

Despite the tight correspondence between the CKO and male C-rescue mice in anxiety, tremor, and acoustic startle response, not all behaviors in the two lines were complementary. For instance, the 6-week-old CKO mice, C-rescue, and stop-null mice all exhibited ataxia, as indicated by a shorter latency to fall from the accelerating rotarod (*Figure 6A and B*). It is understandable that glutamatergic neurons would be necessary but not sufficient to ensure motor coordination, given that multiple neural populations and glia cells are necessary to keep the circuit intact. Strikingly, restoration of MeCP2 only in glutamatergic neurons affected behaviors that did not appear as deficits in the CKO mice but did develop in the stop-null mice, such as social interaction and repetitive behaviors. The CKO mice showed normal interest in familiar and non-familiar partners in the partition test of social behavior (*Figure 6C*), based on the time they spent interacting with partners, and they displayed no repetitive behaviors on the hole-board assay (*Figure 6E*). The stop-null mice, however, showed increased interest in partners and repetitive behavior, both of which were normalized to WT level in the male C-rescue mice (*Figure 6D and F*). It is possible that the rescue of social behavior and stereotypies results from compensatory mechanisms elicited by re-expression of MeCP2 in excitatory neurons.

## Restoration of MeCP2 in glutamatergic neurons in female *Mecp2*-heterozygous mice rescues many RTT-like features

Our behavioral characterization of male C-rescue mice suggests that there is a benefit from restoring glutamatergic function in RTT mouse models, especially on longevity, obesity, anxiety-like behaviors, tremor, repetitive behaviors, and social interaction. The male mice are not the best model to study rescue effects, however, because most RTT patients are females with mosaic expression of MeCP2 due to random X chromosome inactivation. Female *Mecp2*-heterozygous mice are thus a more clinically relevant model for studying the effects of restoring MeCP2 only in glutamatergic neurons. We characterized the female F1 generation from the *Mecp2*$^{LSL/+}$ and *Vglut2*-Cre$^{+/-}$ mating. In these F1 females, *Mecp2*$^{LSL/+}$ (referred as 'stop-het') functionally corresponds to *Mecp2*$^{+/-}$ and *Mecp2*$^{LSL/+}$; *Vglut2*-Cre$^{+/-}$ is the female C-rescue.

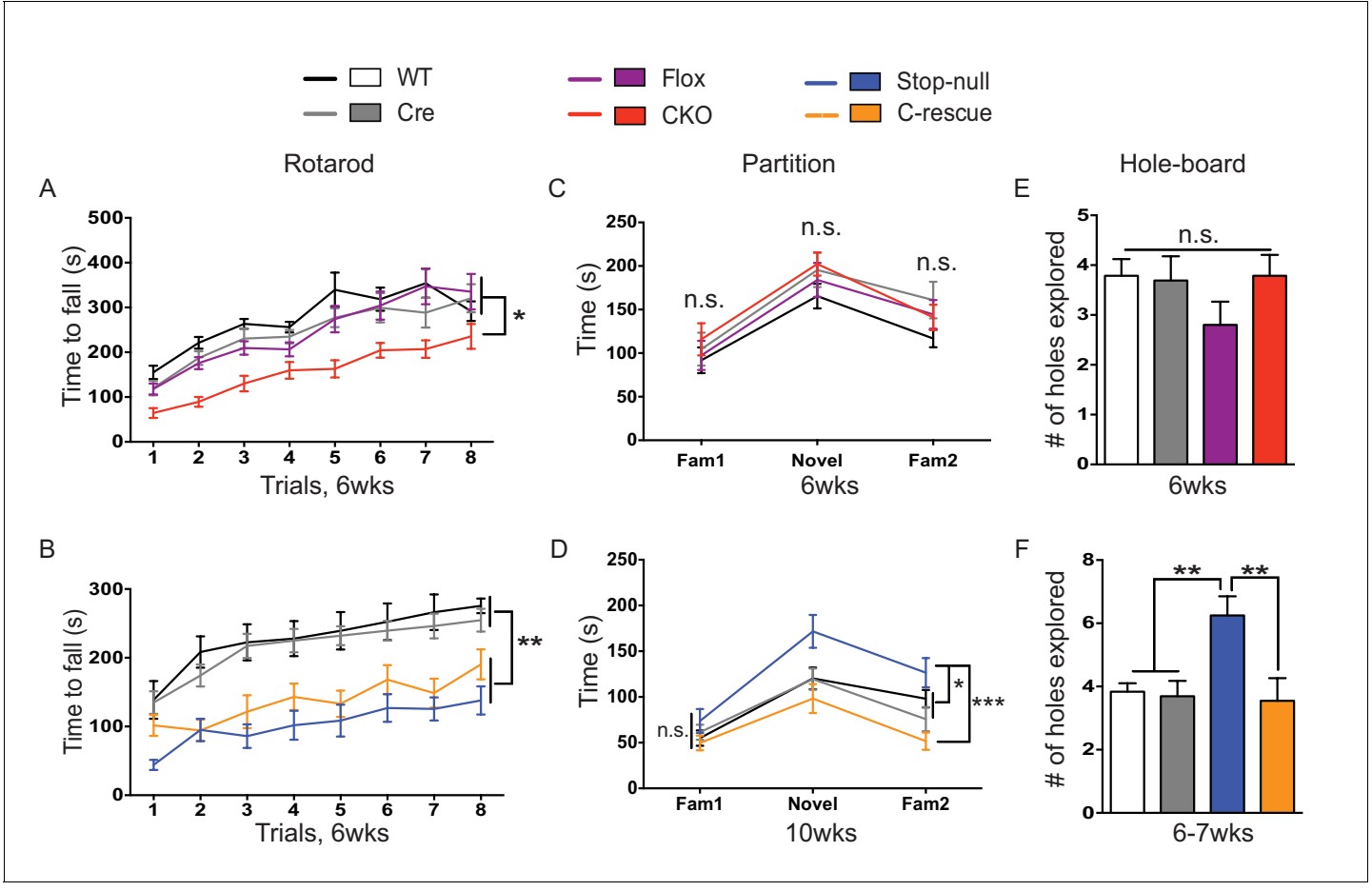

**Figure 6.** Both CKO and male C-rescue mice suffered from ataxia but maintained normal social interaction and were free of repetitive behaviors. (A–B) Latency to fall from the accelerating rotarod plotted as a function of trial (A, n = 15–17; B, n = 9–14). (C–D) Average time that mice spent interacting with partners when they were housed with a familiar mouse (Fam1), a novel mouse (Novel), and the same familiar mouse (Fam2) again in the partition test (C, n = 10–16; D, n = 13–15). (E–F) Number of holes explored with ≥2 sequential nose-pokes during the 10 min hole-board assay (E, n = 10–14; F, n = 11–13). Data are presented as mean ± SEM. *p<0.05; **p<0.01; ***p<0.001; n.s., not significant; by one-way or two-way ANOVA.

Stop-het mice were significantly overweight by 9 weeks of age, while female C-rescue mice, like the male C-rescue, gained much less weight by 10 weeks of age than controls (*Figure 7A*). Similar to *Mecp2*-null mice, the stop-het mice at 10 weeks of age displayed reduced anxiety in the elevated plus maze (*Figure 7B*), and impaired acoustic startle response (*Figure 7C*). Unlike the stop-null mice, the stop-het mice were responsive to acoustic stimulus, so we were able to further analyze the pre-pulse inhibition (PPI). The stop-het mice showed increased inhibition at 78 and 82 dB pre-pulses (*Figure 7D*). Female C-rescue mice performed similarly to controls in all these assays (*Figure 7B–D*).

We then characterized 30-week-old animals to see if the improvements in certain features in female C-rescue mice persisted. All animals still displayed similar locomotor activity at 30 weeks of age (*Figure 7—figure supplement 1A*). Anxiety-like behaviors in stop-het mice at this age were not obvious (*Figure 7—figure supplement 1B*). Correction of impaired acoustic startle response and increased PPI were maintained in female C-rescue mice at 30 weeks of age (*Figure 7—figure supplement 1C and D*). Unexpectedly, in contrast to the male C-rescue mice, female C-rescue mice showed alleviation of ataxia in their significantly longer latency to fall compared to stop-het mice in the rotarod assay (*Figure 7E*). This improvement was further confirmed by fewer footfalls in the foot-slip assay (*Figure 7F*), which assesses motor coordination independent of body weight. In addition, at 30 weeks of age only one out of 28 (3.6%) female C-rescue mice displayed tremor, in contrast to the ~50% stop-het mice that showed tremor (*Figure 7G*). Taken together, our data indicate that

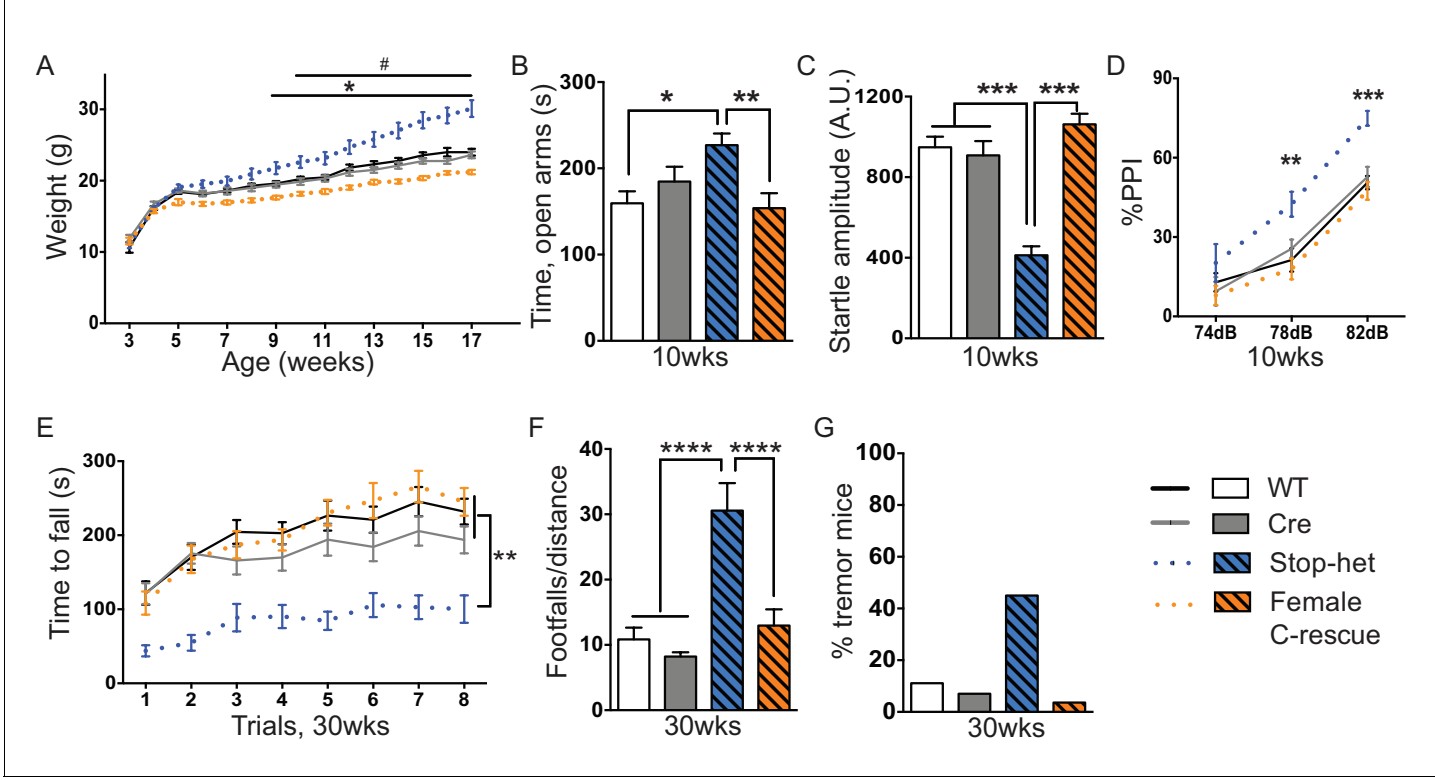

**Figure 7.** Restoration of *Mecp2* in glutamatergic neurons in stop-het mice rescued RTT-like features. (A) Plot of weight as a function of age (n = 16). '*' indicates a statistically significant difference between stop-het and all other genotypes, while '#' indicates a statistically significant difference between female C-rescue and controls. (B) Average time spent in the open arms of the elevated plus maze (n = 16–18). (C) Mean of response to the 120 dB stimulus (n = 16–18). (D) Pre-pulse inhibition at 74 dB, 78 dB and 82 dB pre-pulses (n = 16–18). (E) Latency to fall from the accelerating rotarod is plotted as a function of trial (n = 15–16). (F) Average number of footfalls normalized by distance traveled during the 10 min foot-slip assay (n = 15–16). (G) Percentage of mice displayed tremor at 30 weeks of age (n = 20–28). Data are presented as mean ± SEM. *, # p<0.05; **p<0.01; ***p<0.001; ****p<0.0001; by one-way or two-way ANOVA.

The following figure supplement is available for figure 7:

**Figure supplement 1.** Reversal of impaired acoustic startle and increased PPI were maintained at 30-week-old female C-rescue mice.

restoration of MeCP2 function in glutamatergic neurons in a female heterozygous background rescues many RTT-like features.

## Discussion

As RTT affects virtually every part of the brain and various MeCP2 models have been so thoroughly characterized, MeCP2 mice are an ideal model to interrogate the consequences of partial impairment of a specific neuronal population. This is particularly necessary to postnatal neurological disorders such as ASDs and intellectual disability in which the neurons are present and only partially impaired. Here we have delineated the contributions of MeCP2 loss in glutamatergic neurons to several neurological deficits shared by RTT and ASDs.

Generally excitatory neurons are thought to carry information flow while inhibitory neurons are required for tuning the circuits. Given the entangled nature of excitatory and inhibitory signaling, it is surprising that deleting or restoring MeCP2 only in glutamatergic neurons leads to complementary phenotypes of tremor, anxiety, and acoustic startle response (*Table 1*). In the case of acoustic startle response, this is not entirely unexpected: the caudal pontine reticular nucleus (PnC) is the major sensorimotor interface, and proper glutamatergic signaling is necessary for the transmission of auditory information onto PnC neurons through action on AMPAR (*Koch and Schnitzler, 1997*). As for anxiety and tremor, our findings suggest that the circuits involved are more resistant to inhibitory but

**Table 1.** Comparison of stop-null, *Mecp2* CKO, and male and female C-rescue mice.

| Neurological features | Stop-null | *Mecp2* CKO | Male C-rescue | Female C-rescue |
|---|---|---|---|---|
| Tremor | + | + | - | - |
| Altered Anxiety | + | + | - | - |
| Impaired Acoustic Startle Response | + | + | - | - |
| Premature Lethality | + | + | Significantly delayed | -* |
| Obesity | + | + | - | - |
| Ataxia | + | + | + | - |
| Repetitive Behavior | + | - | - | NE |
| Abnormal Social Interaction | + | - | - | NE |

+: Presence of feature.

-: Absence of feature

*: Deficits were absent in stop-het mice.

NE: Mice were not examined with this particular assay.

not excitatory disturbance, possibly due to their specific connectivity. Similarly, the cerebellar circuits may be more vulnerable to inhibitory alteration since inhibitory Purkinje cells are the sole output of the cerebellar cortex. It is therefore understandable that re-expression of MeCP2 in inhibitory neurons (see companion paper *Ure et al., 2016*) but not excitatory neurons was sufficient to rescue motor coordination deficits. It is also not surprising that several other *Mecp2*-null features, including ataxia, premature lethality, and obesity, can be reproduced in both excitatory and inhibitory *Mecp2* CKO mice, since those circuits are disturbed with alteration in either excitation or inhibition. Thus, restoring MeCP2 in neither excitatory nor inhibitory neurons alone is sufficient to fully rescue premature lethality. Similarly, re-expressing MeCP2 in excitatory neurons failed to maintain body weight, as the C-rescue mice were underweight when compared with WT animals, indicating the necessity of MeCP2 in inhibitory neurons to balance the circuit.

Interestingly, the increased social interaction and repetitive behaviors observed in null mice were not induced in the CKO mice but were improved in the male C-rescue mice. It is worth noting that the GABAergic knockout also shows these defects (*Chao et al., 2010*); it may be that restoring glutamatergic neuron function in the stop-null mice improved GABAergic neuron function through reciprocal interactions.

One intriguing finding of this study is that restoring MeCP2 in glutamatergic neurons benefits stop-het females more than stop-null male mice: ataxia was normalized in the female, but not in the male, C-rescue mice. This is reasonable considering that in addition to the MeCP2 expressed in glutamatergic neurons, female C-rescue mice also retain MeCP2 in over 50% of non-glutamatergic neurons (*Young and Zoghbi, 2004*), none of which express MeCP2 at all in male C-rescue mice. In this regard, it is interesting to note that re-expressing MeCP2 only in GABAergic neurons is less effective in a heterozygous background than in the *Mecp2*-null background (see companion paper *Ure et al., 2016*). The opposite outcomes of these rescue strategies suggest that various types of neurons may be differentially sensitive to MeCP2 dosage, and it seems easier for the brain to compensate for the loss of MeCP2 in 50% of inhibitory neurons (glutamatergic female C-rescue) than in 50% of excitatory neurons (inhibitory female C-rescue). The possible reasons for this difference might include the distinct roles that these two types of neurons play in the brain circuit and the differential changes in gene expression due to loss of MeCP2 in these two types of neurons (*Kodama et al., 2012*; *Telfeian et al., 2003*). Nevertheless, it strongly argues for the use of *Mecp2*-heterozygous mice as the primary RTT disease model when developing and testing therapeutic strategies.

At the circuit level, we demonstrated that deleting MeCP2 only in glutamatergic neurons resulted in reduced cortical activity in layer V pyramidal neurons of the somatosensory cortex, a defect also seen in the *Mecp2*-null mice. Together with the complete rescue of this defect by restoring MeCP2 only in glutamatergic neurons, this indicates an essential role of MeCP2 in modulating the firing activity of excitatory neurons. The dramatically reduced sEPSC charge reflects weakened excitatory drive onto the layer V neurons of CKO mice. Given that neurons usually need to receive above-

threshold excitation to generate action potential (*Hodgkin and Huxley, 1990*), the layer V neurons of CKO mice thus fired less often than controls even though they also received less inhibition. It is worth noting that this reduced firing activity in CKO mice may be region specific, as we also observed seizure-like spike-and-wave discharges in the cortex of these mice. Behavioral or electrographic seizures have also been reported in *Mecp2* whole brain (*Chao et al., 2010*) and forebrain (*Goffin et al., 2014*) inhibitory CKO mice, as well as in somatostatin CKO mice (*Ito-Ishida et al., 2015*), although one group reported absence seizure (*Zhang et al., 2014*) but another group found no behavioral or electrographic seizures at all (*Goffin et al., 2014*) with the forebrain excitatory neuron KO mice. These data suggest that the seizures we observe in *Mecp2*-null mice are caused by multi-level disruption of circuits that can arise from dysfunction of several different neuronal cell types. This is further supported by the C-rescue mice, as restoring MeCP2 only in excitatory neurons is insufficient to prevent seizure.

Previous work has shown that postnatal loss of MeCP2 in forebrain excitatory neurons leads to motor coordination deficits, altered anxiety, and impaired social interaction (*Gemelli et al., 2006*). Through comparing and contrasting the excitatory conditional deletion and conditional rescue models, we confirmed the importance of MeCP2 in glutamatergic neurons to prevent anxiety-like behaviors. Importantly, we also highlighted its contribution to the pathogenesis of tremor, which is a common and highly penetrant symptom in individuals with RTT (*Klauck et al., 2002*; *Roze et al., 2007*), and has not been reported in any other cell type-specific *Mecp2* conditional deletion mouse. Expanding on the observations from Gemelli et al. and on previous studies on the autonomous dysfunction of RTT (*Julu et al., 2001*; *Ward et al., 2011*; *Weese-Mayer et al., 2006*), our work suggests that the autonomic dysfunction and premature death of *Mecp2*-null mice may be mainly driven by impaired excitatory signaling in the brainstem and spinal cord. Our results precisely complement previous knowledge and work from Ure et al., which shows that deficiency of MeCP2 in GABAergic neurons leads to ataxia, increased social interaction, repetitive behaviors, premature lethality, and obesity, but is less involved in tremor, altered anxiety, and impaired acoustic startle response. This work also provides an example of how primary deficits in excitatory signaling lead to neurological deficits, which has ramifications for other postnatal neurological disorders, with disturbances in excitatory and inhibitory homeostasis, such as ASDs and intellectual disability.

## Materials and methods

### Animals

The *Vglut2-Cre*$^{+/-}$ knock-in line was a gift from Dr. Brad Lowell and was backcrossed to the FVB strain for four generations to generate progeny with more than 99% FVB strain polymorphic markers. *Mecp2* CKO mice were obtained by breeding *Mecp2*$^{flox/+}$ (*Guy et al., 2001*) female mice on the 129S6SvEv strain to *Vglut2-Cre*$^{+/-}$ male mice. *Mecp2* conditional rescue mice were obtained by breeding *Mecp2*$^{LSL/+}$ (*Guy et al., 2007*) female mice on the 129S6SvEv strain to *Vglut2-Cre*$^{+/-}$ male mice. CKO mice were compared to wild type, *Mecp2*$^{flox}$, and *Vglut2-Cre* littermate controls. C--rescue mice were compared to wild type, *Vglut2-Cre*, and *Mecp2*$^{LSL}$ littermate controls. Mice were housed in an AAALAS-certified animal facility on a 14 hr/10 hr light/dark cycle. All procedures to maintain and use these mice were approved by the Institutional Animal Care and Use committee for Baylor College of Medicine.

### Behavioral assays

All the behavioral assays were carried out blinded to the genotype. Mice were habituated in the testing room for at least 30 min before the test.

### Open field assay

After habituation in the testing room (200-lux, 60 dB white noise) mice were individually placed in the center of an open Plexiglas chamber (40 × 40 × 30 cm) with photo beams (Accuscan) to measure their activity for 30 min. Data are shown as mean ± standard error of mean and analyzed by one-way ANOVA with Tukey's post hoc analysis.

## Light/dark box

The light/dark box was a chamber with a lit side (36 × 20 × 26 cm) and a dark side (15.5 × 20 × 26 cm) with a 10.5 × 5 cm opening (OmniTech Electronics). After habituation to the testing room (200-lux, 60 dB white noise), the mouse was placed in the lit side. The amount of time animals spent in each side and the number of transitions between the two sides were recorded by photo beams (Fusion) for a 10-minute period. Data are shown as mean ± standard error of mean and analyzed by one-way ANOVA with Tukey's post hoc analysis.

## Elevated plus maze

After habituation to the testing room (200-lux, 60 dB white noise), the mouse was placed in the center of a four-arm maze (each arm 25 × 7.5 cm), with two opposing arms enclosed by 15 cm high walls and the other two open. The maze was 50 cm above the ground level. The amount of time animals spent in, and their entries to, each arm were recorded for 10 min with a camera and ANY-maze (Stoelting Co.) video tracking software. Data are shown as mean ± standard error of mean and analyzed by one-way ANOVA with Tukey's post hoc analysis.

## Acoustic startle response and pre-pulse inhibition

Mice were habituated in a room next to the testing room for 30 min. Test mice were brought into the testing room and were placed in a Plexiglas tube inside of a sound-insulated lighted box (SR-Lab, San Diego Instruments). The procedure was carried out as described previously (*Chao et al., 2010*). Startle stimulus is 120 dB and three pre-pulses used are 74, 78, and 82 dB. Pre-pulse inhibition was calculated as 1-[averaged startle response to startle stimulus with pre-pulse/averaged response to startle stimulus] x 100. Data are shown as mean ± standard error of mean. ASR data are analyzed by one-way ANOVA with Tukey's post hoc analysis. PPI data are analyzed by two-way ANOVA with Tukey's post hoc analysis.

## Accelerating rotarod

Mice were put on an accelerating rotarod (Ugo Basile) whose speed increased from 4 to 40 rpm for the first 5 min and was then maintained at 40 rpm for another 5 min (or was stopped when all the mice fell). Each animal was tested in 4 trials per day for 2 consecutive days, with a 30-minute interval between two trials in the same day. Latency to fall was recorded when the mouse fell from the rod or when the mouse had ridden the rotating rod for two revolutions without regaining control. Data are shown as mean ± standard error of mean and analyzed by two-way ANOVA with Tukey's post hoc analysis.

## Foot-slip test

The apparatus was a floor of parallel-positioned rods within a Plexiglas chamber (21 × 21 cm). Mice were individually placed in the center of the grid. The number of foot slips and the distance traveled were recorded for 10 min with a camera and ANY-maze (Stoelting Co.) video tracking software. The number of footslips was normalized to the total distance traveled in the chamber. Data are shown as mean ± standard error of mean and analyzed by one-way ANOVA with Tukey's post hoc analysis.

## Partition test

This assay was performed as previously described (*Chao et al., 2010*) with a few modifications. Experimental mice were singly housed for 24 hr in standard cages with transparent perforated Plexiglas barrier that separates the cage into two compartments. An age- and gender-matched C57BL/6J partner was placed in the opposite side of the partition cage on the second day. Social interaction scoring was carried out on the third day allowing at least 18 hr co-housing of the experimental and partner mice. The amount of time the experimental mouse exhibited direct interest into the partner in a 5-minute period was recorded using a handheld computer (Psion), and analyzed using the Observer program (Noldus). Three different interaction paradigms were assessed sequentially: experimental mouse versus familiar partner (Familiar 1), experimental mouse versus novel partner (Novel), and experimental mouse versus familiar partner again (Familiar2). Data are shown as mean ± standard error of mean and analyzed by two-way ANOVA with Tukey's post hoc analysis.

## Hole-board assay

This assay was performed as previously described (*Chao et al., 2010*). Briefly, mice were placed on a Plexiglas frame (40 × 40 cm) with 16 equally spaced holes. Mice were allowed to explore freely for 10 min while the experimenter recorded holes that had been nose-poked by the mouse in order. Increased tendency to continuously poke the same hole no less than twice is used as an indicator of repetitive behaviors. Data are shown as mean ± standard error of mean and analyzed by one-way ANOVA with Tukey's post hoc analysis.

## Tremor assessment

Mice were put on the flat palm of the hand for 30 s for a person who is blind to the genotype to decide if the mouse displayed tremor.

## EEG surgery and recordings

The methods were modified from our previous publication (*Hao et al., 2015*). Adult mice were anesthetized with isoflurane. Under aseptic condition, each mouse was surgically implanted with 4 recordings electrodes. Two silver wire electrodes (127 μm diameter) were implanted in the right frontal cortex and somatosensory cortex, respectively, with the reference electrode positioned in the occipital region of the skull. Two tungsten electrodes (50 μm diameter) were aimed at the left hippocampal CA1 (P2.0R1.2H1.3) and dentate (P2.0R1.8H1.8) regions, respectively, with the reference electrode in the corpus callosum. All electrode wires were attached to a miniature connector (Harwin Connector) secured on the skull by dental cement. After 2 weeks of post-surgical recovery, EEG activities (filtered between 0.1 Hz and 1 kHz, sampled at 2 kHz) and the mouse behavior were recorded for 2 hr per day over 3–4 days.

## EEG data analysis

Seizure candidates were identified using custom-written algorithms. Briefly, EEG signals were divided into 10-minute segments. A third order Butterworth bandpass filter of 0.5 and 400 Hz cut-offs was applied to each segment. The filtered data was divided into 500 ms non-overlapping epochs. Signal changes occurring in the time domain were captured by amplitude correlation (auto-correlation value between successive epochs), root mean square (average amplitude of the epoch), and spike density (number of spikes normalized to the epoch). Signal changes occurring in the frequency domain were captured by frequency band ratio, where the power of the upper band (20–50 Hz) was contrasted with that of the lower band (0.5–20 Hz). EEG signals that exceeded the threshold for all of the above quantitative features were identified as seizure candidates. Each candidate and the corresponding video were visually inspected to identify electrographic seizures. All computation was performed in MATLAB R2015b.

## Body composition and indirect calorimetry

This assay was performed as previously described with a few modifications (*Ma et al., 2011*). Mice were group housed before the experiment started. Then they were placed individually in calorimetry chambers for 6 days, with the first 2 days considered as an acclimation phase. Mice had free access to food and water in the first 5 days. Resting metabolic rate (RMR) was measured on day 6 in the Comprehensive Laboratory Animal Monitoring System (CLAMS, Columbus Instruments). Feeders were closed at 6 a.m. on day 6 to prevent any further food consumption. The RMR for each mouse was calculated from the two lowest values of energy expenditure from 4 to 8 hr after the start of the fast. Fat and lean mass were measured by quantitative magnetic resonance spectroscopy (EchoMRI, Houston, TX). For food intake, weight and fat gain, energy expenditure, and resting metabolic rate data were analyzed by ANCOVA with Tukey's post hoc analysis using body weight (for food intake) or fat and lean mass (for energy expenditure) as covariance (*Tschop et al., 2012*). Values are shown as estimated marginal means (least square means) adjusting for covariance ± standard error of mean. For total activity and RER, data are shown as mean ± standard error of mean and analyzed by one-way ANOVA with Tukey's post hoc analysis.

## Auditory brainstem response (ABR) measurements

Auditory brainstem responses (ABRs) were measured as previously described (*Cai et al., 2013*). Briefly, 6-week-old mice were anesthetized with an intraperitoneal injection of ketamine (100 mg/kg) and xylazine (10 mg/kg). Normal body temperature was maintained throughout the procedure by placing the mice on a heating pad. Pure tone stimuli from 4 kHz to 48 kHz were generated using Tucker Davis Technologies System 3 digital signal processing hardware and software (Tucker Davis Technologies, Alachua, FL, USA), and the intensity of the tone stimuli was calibrated using a type 4938 1/4″ pressure-field calibration microphone (Brüel and Kjær, Nærum, Denmark).

Response signals were recorded with subcutaneous needle electrodes inserted at the vertex of the scalp, the postauricular region (reference) and the back leg (ground). Auditory thresholds were determined by decreasing the sound intensity of each stimulus from 90 dB to 10 dB in 5 dB steps until the lowest sound intensity with reproducible and recognizable waves in the response was reached. Data are shown as mean ± standard error of mean and analyzed by two-way ANOVA.

## Cortical slice electrophysiology

Coronal slices (350 µm thick) containing primary somatosensory cortex (S1) were prepared from 6- to 8-week-old mice with a vibratome slicer (Leica Microsystems Inc., Buffalo Grove, IL). Acute slices were collected in chilled (2–5°C) cutting solution containing (in mM) 110 choline-chloride, 25 $NaHCO_3$, 25 D-glucose, 11.6 sodium ascorbate, 7 $MgSO_4$, 3.1 sodium pyruvate, 2.5 KCl, 1.25 $NaH_2PO_4$, and 0.5 $CaCl_2$. Then, slices were incubated in standard artificial cerebrospinal fluid (ACSF, in mM) containing 119 NaCl, 26.2 $NaHCO_3$, 11 D-glucose, 3 KCl, 2 $CaCl_2$, 1 $MgSO_4$, and 1.25 $NaH_2PO_4$ at 37°C for 50 min before being stored at room temperature for recording. All solutions were saturated with 95% $O_2$ and 5% $CO_2$.

Whole-cell recordings were made from visually identified pyramidal neurons in layer 5 region of S1 by using a patch clamp amplifier (MultiClamp 700 B, Molecular Devices, Union City, CA). Micro-electrodes with resistance of 2–3 MΩ were pulled from borosilicate glass capillaries (Sutter Instruments, Novato, CA). The intrapipette solution for recording spontaneous firing (holding at -60 mV) and mEPSC (holding at -70 mV) contained (in mM) 140 potassium gluconate, 5 KCl, 10 HEPES, 0.2 EGTA, 2 $MgCl_2$, 4 MgATP, 0.3 $Na_2$GTP, and 10 $Na_2$-phosphocreatine (pH 7.2). Picrotoxin (100 µM), D-2-amino-5-phosphonopentanoic acid (AP5, 50 µM), and Tetrodotoxin (TTX, 0.1 µM) were present in the ACSF for mEPSC recording. For measuring mIPSCs, the pipettes were filled with (in mM) 145 KCl, 10 HEPES, 2 $MgCl_2$, 4 MgATP, 0.3 $Na_2$GTP, and 10 $Na_2$-phosphocreatine (pH 7.2). AP5 (50 µM), 6-cyano-7-nitroquinoxaline-2, 3-dione (CNQX, 20 µM), and TTX (0.1 µM) were also present in the ACSF. The recordings of spontaneous firing and spontaneous synaptic activity were performed in modified ACSF (in mM: 126 NaCl, 25 $NaHCO_3$, 14 D-glucose, 3.5 KCl, 1 $CaCl_2$, 0.5 $MgCl_2$, 1 $NaH_2PO_4$). For measuring sEPSC/sIPSC, the intrapipette solution contained (in mM) 120 $CsCH_3SO_3$, 20 HEPES, 0.4 EGTA, 5 TEA-Cl (tetraethylammonium chloride), 2 $MgCl_2$, 2.5 MgATP, 0.3 GTP, 10 $Na_2$-phosphocreatine, and 1 QX-314 [N-(2,6-dimethylphenylcarbamoylmethyl) triethylammonium bromide] (pH 7.2 with CsOH). To find the reversal potential, the postsynaptic neurons were voltage-clamped at different potentials (with 5 mV interval) between-75 and -55 mV for sEPSC and between -5 and +20 mVs for sIPSC.

To test the intrinsic property, synaptic blockers (APV 50 µM, CNQX 20 µM, and bicuculine 20 µM) were added into the standard ACSF. The input (current pulse)/output (spikes) curves were generated using incremental depolarizing current pulses (10pA/800 ms).

The whole-cell recording was performed at (30 ± 1°C) by using an automatic temperature controller (Warner Instrument, Hamden, CT). Data acquisition was performed by using a digitizer (DigiData 1440A, Molecular Devices). Minianalysis 6.0.3 (Synaptosoft Inc) and pClamp 10 (Molecular Devices) were used for data analysis. Data were discarded when the rest membrane potential was above −60 mV or the resistance change was over 20% during experiment. Data are shown as mean ± standard error of mean and analyzed by one-way ANOVA with Tukey's post hoc analysis.

## Immunostaining

Experiments was performed as previously described (*Heckman et al., 2014*). Free-floating 45 µm sagittal sections were cut on a cryostat (Leica CM3050 S), and stained overnight with primary antibody: rabbit anti-MeCP2 (Cell Signaling 3456S); mouse anti-MeCP2 (Sigma M6818), rabbit anti-

CamKII (Abcam ab52476), and 2 hr with secondary antibody: goat anti-rabbit Alexa488 (Thermo Fisher Scientific A11034); goat anti-mouse Alexa555 (Thermo Fisher Scientific A-21137). DAPI (Thermo Fisher Scientific D1306) was used to stain nuclei. Sections were mounted with ProLong Gold Antifade Mountant (Thermo Fisher Scientific P36934). Images were acquired on Zeiss 710 laser-scanning confocal microscope and Leica SP8 microscope.

## Statistical analysis

Sample sizes for all analyses were determined by previous experience (*Chao et al., 2010*; *Samaco et al., 2013*). Data were analyzed using commercially available statistical software (Prism6, and SPSS 20.2).

# Acknowledgements

We thank V Brandt, K Ure, A Ishida, and L Lombardi for critical review of the manuscript. This work was supported by NIH/NINDS grant 5R01NS057819 (HYZ) and partly by the Neuroconnectivity Core and Neurobehavior Core of IDDRC at Baylor College of Medicine (1U54HD083092). The content is solely the responsibility of the authors and does not necessarily represent the official views of the Eunice Kennedy Shriver National Institute of Child Health & Human Development or the National Institutes of Health. MX was supported by a Whitehall Foundation Research Grant (2015-05-54) and the Curtis Hankamer Basic Research Fund at Baylor College of Medicine. HYZ is an investigator with the Howard Hughes Medical Institute.

# Additional information

### Competing interests

HYZ: Senior Editor, *eLife*. The other authors declare that no competing interests exist.

### Funding

| Funder | Grant reference number | Author |
| --- | --- | --- |
| Whitehall Foundation | Research Grant 2015-05-54 | Mingshan Xue |
| Baylor College of Medicine | Curtis Hankamer Basic Research Fund | Mingshan Xue |
| National Institute of Neurological Disorders and Stroke | 5R01NS057819 | Huda Y Zoghbi |
| Intellectual and Developmental Disabilities Research Center | 1U54 HD083092 | Huda Y Zoghbi |

The funders had no role in study design, data collection and interpretation, or the decision to submit the work for publication.

### Author contributions

XM, Designed the project, Generated mouse models, Performed behavioral assays, Drafted and revised the article, Designed the experiments, Reviewed and interpreted the data; WW, Designed, performed, and analyzed electrophysiological recording experiment, Drafting and revising the article; HL, Designed the project, Interpreted electrophysiological recording data, Revised the article; L-jH, Performed, and analyzed electrophysiological recording experiment, Revised the article; WC, ESC, Analyzed and interpreted EEG data, Wrote the EEG related figure, legend, and method, Acquisition of data; MLF, Designed and supervised the body composition and indirect calorimetry experiment, Acquisition of data, Analysis and interpretation of data, Drafting or revising the article; BT, Designed and performed EEG recording, Drafting or revising the article; JAH, Acquisition of data, Drafting or revising the article; MLS, FAP, Designed, performed ABR measurements with data analysis, Drafting or revising the article; JLN, Acquisition of data, Analysis and interpretation of data, Drafting or revising the article; JT, Designed and supervised EEG recording, wrote the EEG recordings method part, Acquisition of data; MX, Analyzed and interpreted EEG data, wrote the EEG

related figure, legend, and method, Acquisition of data; HYZ, Designed the experiments, Reviewed and interpreted the data, Drafting or revising the article

## Author ORCIDs
Jose A Herrera, http://orcid.org/0000-0003-3808-1769
Huda Y Zoghbi, http://orcid.org/0000-0002-0700-3349

## Ethics
Animal experimentation: Mice were housed in an AAALAS-certified animal facility. All procedures to maintain and use these mice were approved by the Institutional Animal Care and Use committee for Baylor College of Medicine (Animal protocol number AN-1013).

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
