## [Decision Letter]

Thank you for submitting your article "Manipulations of MeCP2 in glutamatergic neurons highlight their contributions to Rett and other neurological disorders" for consideration by *eLife*. Your article has been reviewed by three peer reviewers, and the evaluation has been overseen by a Senior Editor.

The reviewers have discussed the reviews with one another and the Senior Editor has drafted this decision to help you prepare a revised submission.

In this paper and a companion paper, the authors use a variety of genetic strategies to revisit the relative roles of excitatory and inhibitory neuronal expression of MeCP2 in the pathophysiology of Rett Syndrome as modeled in mice. Although this is an area that has been previously studied, the conclusions reached in prior studies have often apparently been at odds and the present pair of studies attempt to bring some clarity by selectively deleting and restoring MeCP2 in glutamatergic neurons (this paper) or selectively restoring MeCP2 in inhibitory neurons (companion paper).

In this manuscript, the authors examine *Mecp2* loss of function in excitatory neurons, and in turn rescue *Mecp2* function in these cells, to demonstrate that *Mecp2* in excitatory neurons regulates distinct behaviors that include suppression of tremor and anxiety. Importantly, these phenotypes largely do not overlap with the phenotypes observed after *Mecp2* loss of function in inhibitory neurons. This thorough work incorporates unique genetic models (including rescue) and associated behavioral and electrophysiological analysis. Before proceeding further with this manuscript the authors should clarify certain aspects of their data and associated conclusions:

Essential revisions:

1) The physiological analysis (Figure 2 and Figure 2—figure supplement 1) analyzes only spontaneous firing and events. The results agree with prior studies in that suggest MeCP2 regulates recurrent excitatory synaptic connections within the cortex and shows that this requires MeCP2 be expressed in those pyramidal neurons. It is disappointing that this analysis was not carried out for the rescue, since it is not clear if MeCP2 expression is sufficient to rescue this phenotype, but we recognize that not all questions can be addressed in this one manuscript. However, the analysis of inhibitory events does not seem adequate given a prior report (Zhang et al. 2014) that loss of MeCP2 from forebrain excitatory neurons causes loss of inhibition and numerous studies show that reduced activity can lead to reduced trophic support of interneurons and reduced inhibition. The present conclusion that the balance is unchanged is based on spontaneous events, but these in turn depend on the excitatory drive and so changes in inhibition cannot be adequately teased apart. Experiments such as measuring evoked inhibition e.g. using ChR2 as done by Scanziani and colleagues to more accurate measure E/I balance, or measure of GABA mini frequency and amplitude and see whether or not this is altered, would unambiguously clarify the authors data and conclusions about the E/I balance. However, we recognize these experiments are experimentally involved and may exceed the scope of the present work: at minimum, the authors should measure GABA mini frequency and amplitude and see whether or not this is altered (see below).

2) Although the authors conduct current clamp recordings to document action potential firing in cortical neurons as well as voltage clamp recordings to document excitatory versus inhibitory synaptic activity, it is important to analyze the miniature EPSCs and IPSCs in order to compare and contrast these findings to the results presented in Figure 5 of the companion paper.

3) The authors do not address whether or not these animals develop epilepsy and have abnormal EEGs, as reported by Zhang et al.

4) There have been previous publications that reported selective knockout of MeCP2 in glutamatergic neurons and neither of those is cited or discussed here (Gemelli et al., 2006, Goffin et al. 2014, He et al. 2014).

5) Stepping back and evaluating the two studies (Meng et al., as well as Ure et al.,) raises the question of how specific loss of function of *Mecp2* in excitatory versus inhibitory neurons have such non-overlapping phenotypes. Within a network excitatory neurons drive inhibitory activity and in turn inhibitory neurons are the ones that regulate excitatory activity. It is rather surprising that the function of these two neuron populations can be disentangled with clear distinctions. Therefore, it is important for the authors to comment on the interplay and overlap between the two animal models in addition to their distinctions.

---

## [Author Response]

Essential revisions:

1) The physiological analysis (Figure 2 and Figure 2—figure supplement 1) analyzes only spontaneous firing and events. The results agree with prior studies in that suggest MeCP2 regulates recurrent excitatory synaptic connections within the cortex and shows that this requires MeCP2 be expressed in those pyramidal neurons. It is disappointing that this analysis was not carried out for the rescue, since it is not clear if MeCP2 expression is sufficient to rescue this phenotype, but we recognize that not all questions can be addressed in this one manuscript. However, the analysis of inhibitory events does not seem adequate given a prior report (Zhang et al. 2014) that loss of MeCP2 from forebrain excitatory neurons causes loss of inhibition and numerous studies show that reduced activity can lead to reduced trophic support of interneurons and reduced inhibition. The present conclusion that the balance is unchanged is based on spontaneous events, but these in turn depend on the excitatory drive and so changes in inhibition cannot be adequately teased apart. Experiments such as measuring evoked inhibition e.g. using ChR2 as done by Scanziani and colleagues to more accurate measure E/I balance, or measure of GABA mini frequency and amplitude and see whether or not this is altered, would unambiguously clarify the authors data and conclusions about the E/I balance. However, we recognize these experiments are experimentally involved and may exceed the scope of the present work: at minimum, the authors should measure GABA mini frequency and amplitude and see whether or not this is altered (see below).

The reviewers raise an important point here: is MeCP2 expression in glutamatergic neurons sufficient to rescue the impaired excitatory synaptic transmission in the cortex? This seems to be the case, as in Figure 2 we showed that it did rescue the decreased firing rate in stop-null mice. To better analyze inhibitory events we measured GABA mini frequency and amplitude in layer V pyramidal neurons as the reviewers suggested (Figure 2—figure supplement 2). In contrast to Zhang et al. we did not see any significant change across genotypes. This differing result from similar mouse models may be due to the expression of MeCP2 in glia, since the *Emx1*-Cre used by Zhang et al. drives Cre expression in both cortical excitatory neurons and glia, while the *Vglut2-*Cre we used in our study only targets glutamatergic neurons. In addition, the different *Mecp2^flox+/y^*mouse lines used in these two studies may also contribute to the difference.

2) Although the authors conduct current clamp recordings to document action potential firing in cortical neurons as well as voltage clamp recordings to document excitatory versus inhibitory synaptic activity, it is important to analyze the miniature EPSCs and IPSCs in order to compare and contrast these findings to the results presented in Figure 5 of the companion paper.

As suggested, we analyzed mEPSC and mIPSC in CKO, male C-rescue mice, and their controls in Figure 2—figure supplement 2 and Figure 2—figure supplement 3. Please also see the subsection “Expression of MeCP2 in glutamatergic neurons plays an essential role in maintaining excitatory neuron activity”.

3) The authors do not address whether or not these animals develop epilepsy and have abnormal EEGs, as reported by Zhang et al.

We have conducted EEG recording on CKO, male C-rescue mice, and their controls as shown in Figure 4. We did see that the animals have electrographic seizures and that the glutamatergic conditional rescue is not sufficient to normalize the EEG. Please also see the subsection “Removing MeCP2 from excitatory neurons leads to seizure-like discharges in cortical EEG recording”.

4) There have been previous publications that reported selective knockout of MeCP2 in glutamatergic neurons and neither of those is cited or discussed here (Gemelli et al., 2006, Goffin et al. 2014, He et al. 2014).

We apologize for the oversight and have discussed work from Gemelli et al. in the last paragraph of the Discussion and work from Goffin et al. in the fifth paragraph of the Discussion. He et al. does not describe behavioral data from the excitatory conditional deletion mouse they generated, so we did not cite their work in the current study.

5) Stepping back and evaluating the two studies (Meng et al., as well as Ure et al.,) raises the question of how specific loss of function of Mecp2 in excitatory versus inhibitory neurons have such non-overlapping phenotypes. Within a network excitatory neurons drive inhibitory activity and in turn inhibitory neurons are the ones that regulate excitatory activity. It is rather surprising that the function of these two neuron populations can be disentangled with clear distinctions. Therefore, it is important for the authors to comment on the interplay and overlap between the two animal models in addition to their distinctions.

The reviewers raise a critical point on how to compare the two models we generated in the two studies. We have revised the Discussion to not only emphasize the differences of the two CKO mice but also comment more on the overlap of these two animal models. We also speculate as to why we observed such non-overlapping features in the two mouse models. Please see the second paragraph of the Discussion.